# Ethnobotanical Survey in Tampolo Forest (Fenoarivo Atsinanana, Northeastern Madagascar)

**Guy E. Onjalalaina** [1,2,3,4], **Carole Sattler** [2], **Maelle B. Razafindravao** [2], **Vincent O. Wanga** [1,3,4,5], **Elijah M. Mkala** [1,3,4,5], **John K. Mwihaki** [1,3,4,5], **Besoa M. R. Ramananirina** [6], **Vololoniaina H. Jeannoda** [6] and **Guangwan Hu** [1,3,4,*]

1   CAS Key Laboratory of Plant Germplasm Enhancement and Specialty Agriculture, Wuhan Botanical Garden, Chinese Academy of Sciences, Wuhan 430074, China; g.onjalalaina@outlook.com (G.E.O.); vincentokelo@gmail.com (V.O.W.); mkala@wbgcas.cn (E.M.M.); mwihakikarichu@gmail.com (J.K.M.)
2   AVERTEM-Association de Valorisation de l'Ethnopharmacologie en Régions Tropicales et Méditerranéennes, 3 rue du Professeur Laguesse, 59000 Lille, France; carolesattler@hotmail.com (C.S.); maellerazafindravao@gmail.com (M.B.R.)
3   University of Chinese Academy of Sciences, Beijing 100049, China
4   Sino-Africa Joint Research Center, Chinese Academy of Sciences, Wuhan 430074, China
5   East African Herbarium, National Museums of Kenya, P. O. Box 451660-0100, Nairobi, Kenya
6   Department of Plant Biology and Ecology, Faculty of Sciences, University of Antananarivo, BP 566, Antananarivo 101, Madagascar; systemamtith@gmail.com (B.M.R.R.); vololoniaina@yahoo.fr (V.H.J.)
*   Correspondence: guangwanhu@wbgcas.cn

**Abstract:** Abstract: BackgroundMadagascar shelters over 14,000 plant species, of which 90% are endemic. Some of the plants are very important for the socio-cultural and economic potential. Tampolo forest, located in the northeastern part of Madagascar, is one of the remnant littoral forests hinged on by the adjacent local communities for their daily livelihood. However, it has considerably shrunk due to anthropogenic activities forming forest patches. Thus, documenting the useful plants in and around the forest is important for understanding the ethnobotany in this area. **Methods:** In this study we (1) collected and identified useful plants utilized by local communities. Voucher specimens were collected following the information given by interviewees, (2) recorded the collection activities and the consumption methods through semi-structured interviews of the local inhabitants, and (3) performed a phytochemical screening to identify the active compounds and the potential healing metabolites of the medicinal plants. **Results:** A total of 65 people between 25 and 75 years old were interviewed. Surveys recorded 123 species used as timber, food, or medicine. Among them, 92 were forest species and 31 were ruderal species. Medicinal plants were mostly used to cure stomach ailments (71%), fever (33.3%), and fatigue (25%) with leaves (68%) being the most used plant part. Phytochemical analyses of 20 endemic medicinal species showed the presence of compounds that could be responsible for the therapeutic effects of the plants. **Conclusions:** Tampolo forest proves to be an important littoral forest highly utilized by the adjacent local communities due to the presence of a high number of useful plants which are mostly endemic to the region. Hence, our investigation assessed the importance of these species in the locality and this can be used for further study on ecology, conservation, and valorization of these species.

**Keywords:** ethnobotany; Africa; littoral forest; traditional knowledge; phytochemical screening

## 1. Introduction

Tropical forests harbor a rich diversity of species which have a high productive and protective natural values [1,2]. They are also a driver of a significant social and economic development as a result of the exploitation of the existing natural resources [2,3]. Additionally, humans also depend on the forests for food, shelter, and medicines [4]. However, there is a rapid loss of tropical forests through deforestation driven by the increased land use change, natural resource overexploitation, and climate change [5].

Madagascar has a remarkable wealth in terms of vegetation and endemic species. The island is composed of a variety of natural environments, which harbor a unique and globally important assemblage of plant species [6]. It is home to over 14,000 plants species, of which 90% are endemic to the region [3,6]. Among the 490 tree genera on the island, 161 are endemic [7]. However, the increasing intense population growth has led to rapid deforestation as land is cleared for agricultural fields and for fuel. The rainforest cover in Madagascar has recorded a gradual decrease from 5,254,306 hectares in 1990 to 4,489,248 hectares in 2005 [5], and a further loss to 4,345,000 hectares in 2013 [8], which translates to one million hectares lost in 15 years.

Tampolo forest is part of the eastern littoral forest remnants of Madagascar which have shrunk considerably due to anthropogenic activities, hence forming forest patches [6]. The adjacent local communities substantially depend on farming and fishing which generally do not generate enough income, hence the improper exploitation of the forest's natural resources to supplement daily incomes. Due to this direct addition into the wellness of the adjacent community, there is a greater risk of extinction of many endemic biota, such as *Daubentonia madagascariensis* Gmelin "Aye-Ayes" (Daubentoniidae) and *Dalbergia baronii* Baker (Fabaceae). This biodiversity loss is greatly propelled by the forestry sector which have, since the colonial period, focused on the wood production potential of the sites, rather than focus on the region's plants and their practical uses through the traditional knowledge of the local culture and people's perspective [9–14].

As rural communities, local people of Tampolo depend on natural resources for their daily livelihood [13,15,16], especially for their healthcare [17]. In many parts of the world, traditional knowledge has always been transferred orally from generation to generation [18]. However, there is a risk of loss of information over the years, hence the importance of gathering them through ethnobotanical studies all over the world [4,19–24]. In terms of traditional cures, despite the lack of written documents, forest medicinal plant species were used to treat various types of diseases. Unlike other parts of the island where works were completed [17,21,25,26], no related works were available for this present area of study. Therefore, this paper is aimed at (a) filling the gap of the previous literature available by documenting the floristic list of useful forest plants with emphasis on medicinal endemic species, (b) reviewing their traditional therapeutic uses, and (c) documenting other uses of the forest plants other than medicinal purposes. Additionally, it evaluates the significance of most salient plant families, genera, and species and their uses among the participants for the conservation of the biological resources and their sustainable utilization.

## 2. Materials and Methods

### 2.1. Study Area

Tampolo forest is located on the eastern coast of Madagascar covering an area of 360 ha (Figure 1). It is about 110 km from Toamasina, the capital district and 10 km from Fenoarivo Atsinanana, in Analanjirofo region [27]. It is bordered by Lake Tampolo and the village of Rantolava to the north, the village of Ampasimazava to the south, the Indian Ocean to the east, and the National Road Number 5 and the village of Tanambao Tampolo to the west. It is classified as low altitude dense evergreen humid forest belonging to the series of *Anthostema* and Myristicaceae by Humbert and Cours-Darne [28] and as coastal forest by DuPuy and Moat [29], recording over 360 plant species [30]. Three types of soils can be found in the forest station: Sandy soils are mainly found on the southern side of the station, whereas on the north and western sides have clay-loamy grounds with hills having ferralitic soils. Tampolo region has an average annual temperature of 23 °C with the coldest month being July with 19 °C, and December is the hottest month with 26.5 °C. The region receives 3406 mm of rain per year with an average of 241 rainy days registered per year. In terms of human population, nearly 6000 people are distributed within the following "fokontany": Andapa II, Tanambao Tampolo, Rantolava, and Takobola, which belong to the rural "commune" of Ampasina Maningory as of 2014.

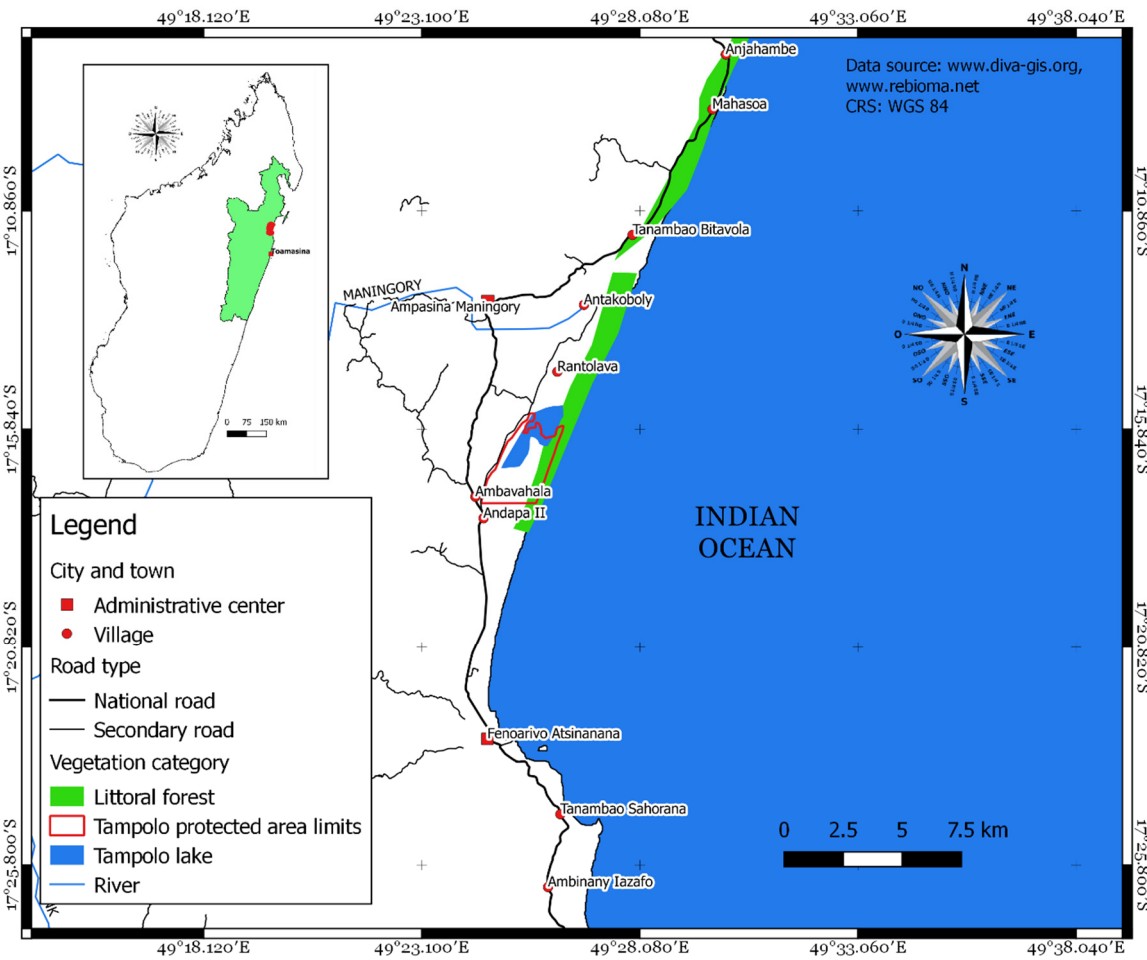

**Figure 1.** Map of the study area: Location of Tampolo forest and the adjacent communities living around the forest.

### 2.2. Ethnobotanical Data Collection

Five field trips for ethnobotanical and biological surveys were carried out from February to November 2012. The participants were randomly selected based on the use and the in-depth knowledge on the medicinal plants. Some of the informants were chosen based on referrals by the local chiefs and elders. Ethnobotanical surveys were done using the methods of Martin [31]. To achieve this, open or semi-open interviews were used, which means that questionnaires (Appendix A), were administered in a manner that could not influence the answers of the participants and by following the International Society of Ethnobiology (ISE) code of ethics [32]. Researchers started their interaction with each prospective respondent by first explaining the aims and objectives of the project in order to solicit their consent and co-operation before any ethnobotanical data were gathered. Interviews were conducted with the selected informants to determine and explore the ethnobotanical knowledge regarding the utilization of plant species, their usefulness, their utilized parts, mode of preparation, or method of processing the plants. The ethnobotanical data for this research were gathered from 53 farmers, three traditional healers, three fishermen, two chiefs of Fokontany, one forest guard, one Tangalamena (traditional chief of the village), one retired nurse, and one blacksmith. The participants were selected based on their consent to be interviewed, their vast knowledge in indigenous species, and their affirmation on the use of or having knowledge of at least one of the uses of the forest plant species. At first, before any interview, we introduced ourselves to those responsible for the village; after their agreement, participants were asked if they can be interviewed in the context of our study. They were free to participate or not. This study was carried out under

the permission and approval of the University of Antananarivo, Faculty of Sciences. No human or animal experiments were carried out.

Investigations were conducted in the three villages surrounding the forest: Tanambao Tampolo, Andapa II, and Rantolava where informants of the age 25 and above were interviewed and with their prior consent. The Nagoya protocol on access and benefit-sharing [33] has been followed. All the methods were performed in accordance with the relevant guidelines and regulations and, informed consent was obtained from all the participants during the period of this study.

The importance value of the use of each species by local population were assessed by calculating its Use Index by using the formula of Lance et al. [34]:

$$I\ (\%) = n/N \times 100 \tag{1}$$

where I (%) is the percentage index of use, n is the number of people citing the species, and N is the total number of people surveyed. A given species is mostly used if the value of I (%) is between 60 and 100% and moderately used if I (%) is between 30 and 60%; and if I (%) is less than 30%, it is rarely used.

### 2.3. Specimen Collection and Taxonimic Identification

Voucher specimens were collected with the help of the field guides and following the information given by interviewees relying on the local names of the plant taxa. Some of the plant species were identified in the field with the help of the locals and the remaining ones identified by Guy Eric Onjalalaina at the herbarium of Tsimbazaza Park (TAN), according to the Index Herbariorum list [35]. The plants were pressed and stored in the Herbarium. The taxonomy of taxa included in this study is consistent with Tropicos, International Plant Name Index (IPNI), the Plants of the World Online (POWO), and Plant List databases [36–39]. Duplicates were deposited at the office of the "Association de Valorisation de l'Ethnopharmacologie en Regions Tropicales et Meditéranéenes (AVERTEM)" in Tampolo and at the herbarium of the "Département de Biologie et Ecologie Végétales (DBEV)" which is not yet listed at the Index Herbariorum, Faculty of Sciences Ankatso. A unique voucher specimen number was assigned to each herbarium specimen.

### 2.4. Phytochemical Screening for Medicinal Plants

2.4.1. Extraction Process by Reflux and Soxhlet Extractions

The leaves of the selected medicinal plant species were pre-washed with distilled water to rinse off the dirt and were dried in a ventilated area under a shade. Reflux and Soxhlet extraction involved distillation processes which are widely used in food and non-food industrial processes and laboratories. The process involves heating a solution to its boiling and then returning the condensed vapor to the original flask. An aqueous extract was prepared by mixing 1 g of leaf powder with 20 mL of distilled water, then the solution was boiled and cooled [40–42].

2.4.2. Quantitative Analyses

The tannins were identified using the ferric chloride test where drops of 1% $FeCl_3$ solution in methanol were added to the 2 mL of hydroalcoholic extract, the blue color indicating the presence of tannins. On the other hand, phenolic compounds were detected when the color of the mixture switches to dark blue or blue-green by mixing four drops of ferric chloride in methanolic solution with 0.5 mL of the extract [42]. Then, anthraquinones were detected using the Bornträger reaction [43,44], 0.5 mL of the aqueous solution was mixed with 1 mL of benzene. After decantation, 0.5 mL of ammoniac 25% was added, the turn to red of the solution indicates the presence of anthraquinones. The presence of desoxyoses were also detected using 0.5 mL of the aqueous solution with, consecutively, 0.5 mL of cold acetic acid, 0.5 mL of ferric chloride 10% and 0.5 mL of sulfuric acid ($H_2SO_4$) 36 N, where "N" is the number of particles in the substance (reaction of Keller-Kiliani [45]). The formation of a purplish ring at the interface of the tube confirms the presence of

desoxyoses [46]. Iridoids were detected by adding some drops of hydrochloric acid (HCl) 12 N to 0.5 mL of the aqueous solution. The mixture was boiled in a water bath for 30 min, then a dark green or dark blue precipitate or color appears if these compounds are presents. For saponins, after dissolving in distilled water, there should be a formation of a foamy solution after strongly shaking for 2 min [40], and if the convoluted foam persisted within 15 min, it contained saponins.

Chloroform extract was used to detect the presence of steroids and terpenes. One gram of leaf powder was mixed with chloroform, stored in a cold place for one night, then filtered. The Libermann–Burchard test [47] was used by mixing 1 mL of the extract with 1 mL of acetic anhydride. After shaking, 1 mL of $H_2SO_4$ was then added. The formation of a purplish red ring indicates the presence of terpenes while the presence of steroids was indicated by the formation of a green color at the upper level of the solution. Additionally, the presence of sterols was detected by using the reaction of Salkowski [48,49], The phase at the bottom of the test tube turns in red if they were presents when 0.5 mL of $H_2SO_4$ 34 N and three drops of anhydrous acetic were added to 0.5 mL of chloroform extract.

After that, 1 g of leaf powder was mixed with 10 mL of hydro ethanol (75%) then stored in a cold place for one night. To detect the presence of flavonoids, the Wilstater procedures [50,51] was used by adding four drops of HCl 12 N and two magnesium turnings to 2 mL of the extract. The color change to red indicates the presence of flavonoid compounds. Then, the detection of anthocyanins followed the procedures of Bate-Smith [52]. A mixture of 2 mL of the plant extract and 0.5 mL of HCl 12 N was boiled for 30 min and, when cooling, a red color appeared.

Finally, 1 g of leaf powder was mixed with 10 mL of HCl 2 N and marinated for one night. Then, 1 mL of the acid extract were then mixed with four drops of reagent of Mayer [53], Wagner [54] or Dragendorff [55] and produced a white precipitate or a flocculation if alkaloids were presents in the solution.

## 3. Results

### 3.1. Demographic Variables

During the study, 65 local inhabitants were surveyed, 41 (63.08%) were male and 24 (36.92%) were female. The age of the informants ranged from 25 and 75 years old (Table 1). The survey was done either by individual interviews (one-on-one consultations) or through focus groups.

**Table 1.** Distribution of the interviewees according to the age group and the gender.

| Age (y) | Male | % Frequency | Female | % Frequency | Total |
|---------|------|-------------|--------|-------------|-------|
| 25–35 | 23 | 56.1 | 8 | 33.3 | 31 |
| 36–45 | 5 | 12.2 | 3 | 12.5 | 8 |
| 46–55 | 3 | 7.3 | 8 | 33.3 | 11 |
| 56–65 | 8 | 19.5 | 4 | 16.7 | 12 |
| 66–75 | 2 | 4.9 | 1 | 4.2 | 3 |
| Total | 41 | 100 | 24 | 100 | 65 |

### 3.2. Plant Utilizations

The following ethnobotanical information are reported for each taxon: the scientific name, the family name, the growth form, the plant part used, and uses. During these interviews, 123 taxa distributed within 62 families and 112 genera, including ruderal species, were cited as useful in the locality of Tampolo of which 59 were medicinal (48%), 54 for timber and firewood (44%) and 10 were edible (8%) (Figure 2). Among these useful plants, 92 taxa are exclusively from the forest, where 78 (84.78%) of them were endemic (Table 2) then distributed within 49 families and 83 genera. Most of the forest-utilized plant families were represented by two or three taxa.

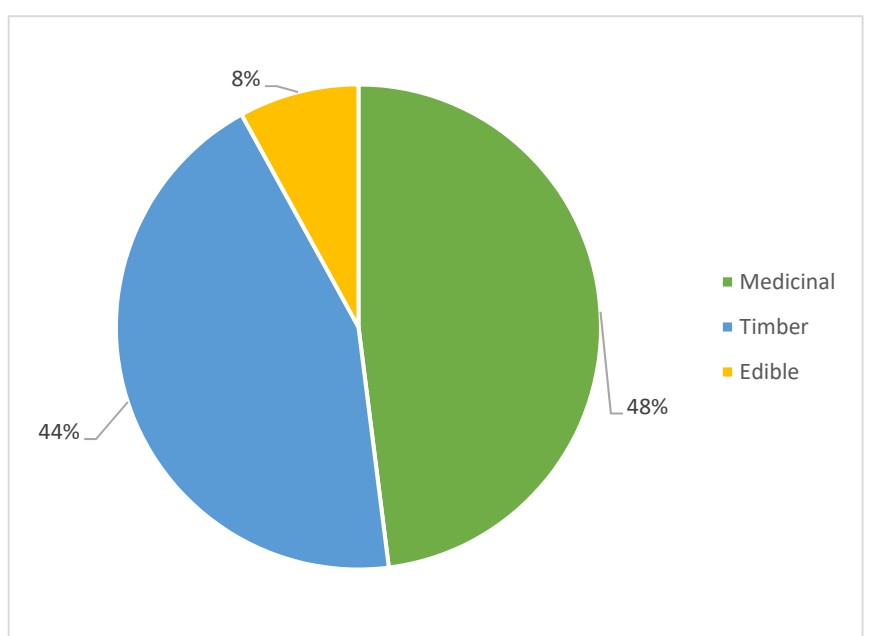

**Figure 2.** Graphic representation of the utilization of the plants.

**Table 2.** List of useful plants of Tampolo forest (Fenoarivo Atsinanana, Madagascar) with their Use Index {I (%)}. M: Medicinal; T: Timber; E: Edible.

| | Family | Taxon | Collection Number | Local Name | Use | Endemism | Life Form | I (%) | IUCN Status |
|---|---|---|---|---|---|---|---|---|---|
| 1 | Anacardiaceae | *Sorindeia madagascariensis* Thouars ex DC. | GE 109 | Voantsirindrina | E | Not endemic | Liana | 27.7 | Not assesed |
| 2 | | *Campnosperma micranteium* Marchand | GE 034 | Tarantana | T | Endemic | Tree | 6.1 | LC |
| 3 | Anisophylleaceae | *Anisophyllea fallax* Scott-Elliot | GE 124 | Hazomamy | M | Endemic | Tree | 9.2 | LC |
| 4 | Annonaceae | *Xylopia buxifolia* Baill | GE 122 | Hazoambo | M | Endemic | Tree | 46.1 | LC |
| 5 | | *Fenerivia ghesquiereana* (Cavaco & Keraudren) R.M.K. Saunders | GE 096 | Tsilongodongotra | T | Endemic | Tree | 6.1 | LC |
| 6 | Apocynaceae | *Landolphia nitens* Lassia | GE 013 | Voahena | E | Endemic | Liana | 46.1 | Not Assessed |
| 7 | | *Tabernaemontana retusa* (Lam.) Pichon | GE 126 | Livoro | M | Endemic | Tree | 9.2 | LC |
| 8 | | *Stephanostegia capuronii* Markgr | GE 110 | Hazon-dronono | T | Endemic | Tree | 9.2 | LC |
| 9 | Araceae | *Pothos scandens* L. | GE 006 | Ravin-tampina | M | Not endemic | Vine | 9.2 | Not Assessed |
| 10 | Araliaceae | *Schefflera vantsilana* (Baker) Bernardi | GE 106 | Voantsilana | T | Endemic | Tree | 3 | LC |
| 11 | Arecaceae | *Dypsis fasciculata* Jum | GE 056 | Amboza | T | Endemic | Shrub | 9.2 | NT |
| 12 | Asclepiadaceae | *Secamone obovata* Decne. | GE 090 | Vahizahana | M,E | Endemic | Vine | 21.5 | LC |
| 13 | Asteropeiaceae | *Asteropeia micraster* Hallier F. | GE 024 | Tambônana | T | Endemic | Tree | 24.6 | VU |
| 14 | | *Asteropeia matrambody* (Capuron) G.E.Schatz, Lowry & A.-E.Wolf | GE 023 | Matrambody | T | Endemic | Tree | 6.1 | VU |
| 15 | Bignoniaceae | *Phyllarthron bojeranum* DC. | GE 093 | Antohiravina | M,T | Endemic | Tree | 27.7 | LC |

**Table 2.** *Cont.*

| | Family | Taxon | Collection Number | Local Name | Use | Endemism | Life Form | I (%) | IUCN Status |
|---|---|---|---|---|---|---|---|---|---|
| 16 | | *Rhodocolea racemosa* (Lam.) H.Perrier | GE 102 | Velonavohitra | T | Endemic | Shrub | 12.3 | Not Assessed |
| 17 | | *Colea tetragona* DC. | GE 042 | Sifontsoy | M | Endemic | Shrub | 6.1 | |
| 18 | Burseraceae | *Aucoumea klaineana* Pierre | GE 123 | Akomea | T | Endemic | Tree | 27.7 | VU |
| 19 | Celastraceae | *Brexia madagascariensis* (Lam.) Thouars ex Ker Gawl. | GE 125 | Maimboholatra | M | Not endemic | Shrub | 9.2 | LC |
| 20 | Clusiaceae | *Symphonia fasciculata* (Noronha ex Thouars) Vesque | GE 112 | Haziny | T | Endemic | Tree | 18.5 | VU |
| 21 | | *Garcinia* sp. | GE 065 | Ravi-masina kakazo | M | Endemic | Shrub | 6.1 | Unknown |
| 22 | | *Calophyllum paniculatum* P.F.Stevens | GE 033 | Vintanona | T | Endemic | Tree | 3 | VU |
| 23 | | *Symphonia* sp. | GE 130 | Haziny be ravina | T | Endemic | Shrub | 1.5 | |
| 24 | Combretaceae | *Terminalia catappa* L. | GE 116 | Antafana | T | Not endemic | Tree | 18.5 | LC |
| 25 | Connaraceae | *Agelaea pentagyna* (Lam.) Baill. | GE 008 | Vahimaintina | M | Not endemic | Liana | 43 | Not Assesed |
| 26 | Dilleniaceae | *Tetracera madagascariensis* Willd. ex Schltdl. | GE 016 | Vahimaragna | M | Endemic | Liana | 12.3 | Not Assesed |
| 27 | | *Hibbertia coriacea* (Pers.) Baill. | GE 067 | Anjavidy vavy | M | Endemic | Shrub | 3 | Not Assessed |
| 28 | Ebenaceae | *Diospyros filipes* H.Perrier | GE 049 | Hazomaintina | T | Endemic | Tree | 12.3 | VU |
| 29 | | *Diospyros* sp. | GE 050 | Hazomaintina | T | Endemic | Epiphyte | 1.5 | Unknown |
| 30 | Elaeocarpaceae | *Elaeocarpus alnifolius* Baker | GE 057 | Aferonakavy | M | Endemic | Shrub | 3 | LC |
| 31 | Ericaceae | *Erica* sp. | GE 059 | Anjavidy lahy | M | Endemic | Shrub | 40 | Unknown |
| 32 | | *Vaccinium* sp. | GE 121 | Voantsirihitra | E | Endemic | Shrub | 15.4 | Unknown |
| 33 | Euphorbiaceae | *Croton noronhae* Baill. | GE 043 | Fotsy avadika | M | Endemic | Shrub | 18.5 | Not Assessed |
| 34 | Fabaceae | *Intsia bijuga* (Colebr.) Kuntze | GE 073 | Hintsina | T | Not endemic | Tree | 86 | NT |
| 35 | | *Dalbergia baronii* Baker | GE 047 | Hazovola | T | Endemic | Tree | 77 | VU |
| 36 | | *Dialium unifoliolatum* Capuron | GE 048 | Zahana (zana) | M | Endemic | Tree | 15.4 | NT |
| 37 | | *Cynometra capuronii* Du Puy & R.Rabev. | GE 046 | Mampay | M | Endemic | Tree | 6 | EN |
| 38 | | *Hymenaea verrucosa* Gaertn. | GE 071 | Mandrofo | T | Not endemic | Tree | 6 | Not Assessed |
| 39 | Gentianaceae | *Tachiadenus carinatus* (Desr.) Griseb. | GE 003 | Rangilo | M | Endemic | Herb | 3 | LC |
| 40 | | *Anthocleista madagascariensis* Baker | GE 064 | Dindemo | M | Endemic | Tree | 1.5 | LC |
| 41 | Hypericaceae | *Psorospermum chionanthifolium* Spach | GE 098 | Harongam-panihy | T | Endemic | Shrub | 3 | Not Assessed |
| 42 | Lauraceae | *Ocotea racemosa* (Danguy) Kosterm. | GE 086 | Tafononana | T | Endemic | Tree | 6.1 | LC |
| 43 | | *Cryptocarya* sp. | GE 045 | Tavolo | T | Endemic | Tree | 6.1 | Unknown |
| 44 | | *Cryptocarya acuminata* Merr. | GE 044 | Tavolomalama | T | Not endemic | Tree | 3 | Not Assessed |
| 45 | Liliaceae | *Dracaena reflexa* Lam. | GE 052 | Felana | T | Not endemic | Shrub | 1.5 | LC |
| 46 | | *Dracaena* sp. | GE 053 | Felana | T | Not endemic | Shrub | 1.5 | Unknown |

**Table 2.** *Cont.*

|  | Family | Taxon | Collection Number | Local Name | Use | Endemism | Life Form | I (%) | IUCN Status |
|---|---|---|---|---|---|---|---|---|---|
| 47 | Melastomataceae | *Medinilla parvifolia* Baker | GE 004 | Ravi-masina | M | Endemic | Epiphyte | 9.2 | Not Assessed |
| 48 |  | *Medinilla quadrangularis Jum. & H. Perrier* | GE 005 | Ravi-masina | M | Endemic | Epiphyte | 1.5 | Not Assessed |
| 49 |  | *Memecylon thouarsianum* Naudin | GE 082 | Tsimahamasatokin | T | Endemic | Tree | 6.1 | EN |
| 50 |  | *Memecylon* sp. | GE 132 | Tsimahamasatokina | T | Endemic | Tree | 1.5 | Unknown |
| 51 | Menispermaceae | *Burasaia madagascariensis* DC. | GE 032 | Hazon-dahy | M | Endemic | Tree | 15.3 | LC |
| 52 |  | *Tinospora* sp. | GE 017 | Andanitrehy | M | Endemic | Liana | 9.2 | Unknown |
| 53 | Monimiaceae | *Tambourissa religiosa* (Tul.) A. DC | GE 114 | Ambora | T | Endemic | Shrub | 3 | LC |
| 54 | Moraceae | *Trilepisium* sp. | GE 028 | Tsopatika | T | Endemic | Tree | 12.3 | Unknown |
| 55 |  | *Streblus dimepate* (Bureau) C.C. Berg | GE 127 | Maherihely | T | Endemic | Tree | 9.2 | LC |
| 56 |  | *Ficus lutea* Vahl | GE 062 | Amontana | M | Not endemic | Tree | 3 | LC |
| 57 | Myristicaceae | *Brochoneura acuminata* (Lam.) Warb. | GE 030 | Rara | M | Endemic | Tree | 15.3 | LC |
| 58 | Myrsinaceae | *Oncostemum botryoides* Baker | GE 088 | Hazontoho | T | Endemic | Shrub | 6 | Not Assessed |
| 59 | Myrtaceae | *Syzygium bernieri* (Baill. ex Drake) Labat & Schatz | GE 113 | Hompa | T | Endemic | Tree | 21.5 | LC |
| 60 | Ochnaceae | *Campylospermum obtusifolium* (DC.) Tiegh. | GE 089 | Menahihy | M | Endemic | Shrub | 9.2 | Not Assessed |
| 61 | Olacaceae | *Olax emirnensis* Baker | GE 087 | Famelondriaka | M | Endemic | Tree | 1.5 | LC |
| 62 | Oleaceae | *Noronhia boivinii* Dubard | GE 084 | Tsilaitra | M | Endemic | Tree | 9.2 | NT |
| 63 |  | *Noronhia* sp. | GE 131 | Tsilaitra be ravina | T | Endemic | Tree | 1.5 | Unknown |
| 64 | Phyllanthaceae | *Bridelia tulasneana* Baill. | GE 009 | Roihavitra | M | Endemic | Tree | 15.3 | LC |
| 65 |  | *Cleistanthus capuronii* Leandri | GE 039 | Lohendry | T | Endemic | Tree | 6.1 | EN |
| 66 |  | *Uapaca thouarsii* Baill. | GE 120 | Voapaka | M,T,E | Endemic | Tree | 98.5 | LC |
| 67 |  | *Wielandia mimosoides* (Baill.) Petra Hoffm. & McPherson | GE 027 | Beando | T | Endemic | Shrub | 3 | LC |
| 68 | Physenaceae | *Physena madagascariensis* Steud. | GE 094 | Fanavimangoaka | M | Endemic | Shrub | 9.2 | LC |
| 69 | Pittosporaceae | *Pittosporum ochrosiifolium* Bojer | GE 095 | Maimbovitsika | M | Endemic | Shrub | 6.1 | LC |
| 70 | Putranjivaceae | *Drypetes madagascariensis* (Lam.) Humbert & Leandri | GE 054 | Tsivavegny | M | Endemic | Shrub | 15.3 | LC |
| 71 | Rhamnaceae | *Bathiorhamnus louvelii* (H.Perrier) Capuron | GE 026 | Menavahatra | M | Endemic | Tree | 1.5 | LC |
| 72 | Rhizophoraceae | *Macarisia pyramidata* Thouars | GE 080 | Hazomalagny | M | Endemic | Tree | 15.3 | LC |
| 73 | Rosaceae | *Magnistipula tamenaka* (Capuron) F.White | GE 128 | Tamenaka | T | Endemic | Tree | 6.1 | LC |
| 74 | Rubiaceae | *Saldinia axillaris* (Lam. ex Poir.) Bremek. | GE 103 | Valavelona | M | Endemic | Shrub | 6.1 | Not Assessed |

**Table 2.** *Cont.*

| | Family | Taxon | Collection Number | Local Name | Use | Endemism | Life Form | I (%) | IUCN Status |
|---|---|---|---|---|---|---|---|---|---|
| 75 | | *Pyrostria media* (A.Rich. ex DC.) Cavaco | GE 101 | Tsifo madini-Dravina | T | Endemic | Shrub | 6.1 | LC |
| 76 | | *Breonia madagascariensis* A.Rich. ex DC. | GE 029 | Molo-pangady | M | Endemic | Tree | 3 | CR |
| 77 | | *Pyrostria major* (A.Rich. ex DC.) Cavaco | GE 100 | Tsifobe | M | Endemic | Tree | 3 | LC |
| 78 | | *Hyperacanthus poivrei* (Drake) Rakotonas. & A.P.Davis | GE 072 | Voantalanina | T | Endemic | Tree | 3 | NT |
| 79 | | *Gaertnera* sp. | GE 064 | Sadôdôka | M | Endemic | Tree | 3 | Unknown |
| 80 | Salicaceae | *Homalium erianthum* (Tul.) Baill. | GE 068 | Hazom-bato | T | Endemic | Tree | 9.2 | VU |
| 81 | | *Ludia madagascariensis* Clos | GE 077 | Fanenton'akoholahy | M | Endemic | Shrub | 3 | LC |
| 82 | Sapindaceae | *Pseudopteris decipiens* Baill. | GE 097 | Hazomananjara | M | Endemic | Shrub | 6.1 | Not Assessed |
| 83 | | *Filicium thouarsianum* (DC.) Capuron | GE 063 | Elatrangidina | T | Endemic | Tree | 3 | NT |
| 84 | Sapotaceae | *Mimusops coriacea* (A.DC.) Miq. | GE 083 | Voaranto | E | Not endemic | Tree | 61.5 | Not Assessed |
| 85 | | *Faucherea glutinosa* Aubrév. | GE 061 | Nanto | O | Endemic | Tree | 46 | Not Assessed |
| 86 | | *Labramia bojeri* A.DC. | GE 074 | Nanto vasihy | T | Endemic | Tree | 6.1 | Not Assessed |
| 87 | | *Chrysophyllum boivinianum* (Pierre) Baehni | GE 038 | Famelona | M | Not endemic | Tree | 3 | LC |
| 88 | Sarcolaenaceae | *Leptolaena abrahamii* G.E.Schatz & Lowry | GE 075 | Amanin'aombilahy | T | Endemic | Tree | 21.5 | NT |
| 89 | | *Schizolaena rosea* Thouars | GE 107 | Tsiariagnarany | T | Endemic | Tree | 9.2 | VU |
| 90 | | *Sarcolaena grandiflora* Thouars | GE 104 | Helana | T | Endemic | Tree | 6 | VU |
| 91 | | *Schizolaena* sp. | GE 108 | Voandroza | T | Endemic | Tree | 6 | Unknown |
| 92 | Simaroubaceae | *Quassia indica* (Gaertn.) Noot. | GE 129 | Bemafaitra | M | Not endemic | Tree | 1.5 | LC |

**Note**: **LC**-Least Concern; **VU**-Vulnerable; **NT**-Near Threatened; **CR**-Critically Endangered; **EN**-Endangered.

### 3.3. Growth Form of the Plants

Trees (56 taxa) and shrubs (25 taxa) were cited by the participants to be the most exploited (Figure 3) while climbers (seven taxa), epiphytes (three taxa), and herbs (one taxon) were least cited. Four species, such as *Uapaca thouarsii* Baill., *Intsia bijuga* Kuntze, *Dalbergia baronii* Baker, and *Mimusops coriacea* Miq., had their use index greater than 60% because they were highly valued by the local people as timber. Moreover, *U. thouarsii* was used as medicinal plant. Few forest species were edible and fruits were the major part that were eaten by the local people. Furthermore, these species were also eaten by lemurs and birds.

### 3.4. Medicinal Plants

Among the useful forest plants, 43 taxa were medicinal having therapeutical values. However, among the 43 taxa, three taxa were also used as timber as well as food, while 37 of them were endemic to Madagascar (Table 2). The most frequent diseases that are treated with plants were diarrhea, stomachache, oral, dental, genital infections, and non-malaria fever (Table 3). In the case of malaria, all of the interviewees affirmed that they

consult a doctor. In particular, leaves were the most frequently used parts (68%) that were used in the cure of most of the diseases as shown in the Figure 4. The other parts or components of the plant such as stem, root, bark, and latex or a mixture of two or more of them were also used in low proportions.

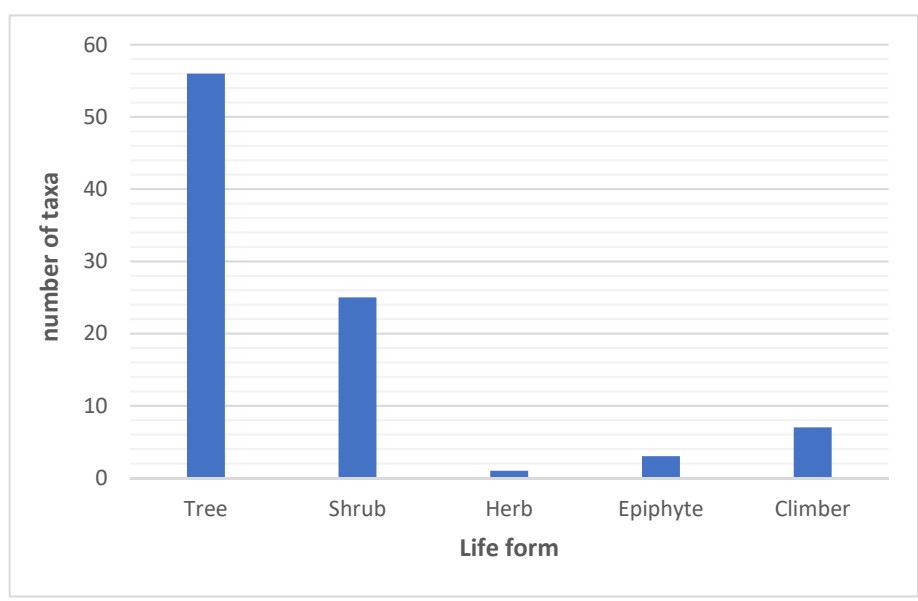

**Figure 3.** Distribution of taxa for each lifeform.

**Table 3.** Method of use of the forest medicinal plants.

| Family | Taxon | Healing Properties | Part Used | Method of Preparation |
|---|---|---|---|---|
| Anisophylleaceae | *Anisophyllea fallax* Scott-Elliot | Anti-fever | Leaf | Decoction |
| Annonaceae | *Xylopia buxifolia* Baill. | Antidiarrheal Anti-fatigue. | Leaf | Decoction |
| Apocynaceae | *Tabernaemontana retusa* (Lam.) Pichon | Against toothache | Latex | Poultice |
| Asclepiadaceae | *Secamone obovata* Decne. | Anti-yellow fever | Leafy branch | Decoction |
| Bignoniaceae | *Colea tetragona* DC. | Anti-genital infections | Leaf | Decoction, infusion |
| | *Phyllarthron bojeranum* DC. | Anti-stomach ache Anti-fatigue | Leaf | Decoction |
| Clusiaceae | *Garcinia* sp. | Anti-prolonged cough for kid | Leaf | Decoction |
| Dilleniaceae | *Hibbertia coriacea* (Pers.) Baill. | Anti-fever | Leafy branch | Decoction |
| | *Tetracera madagascariensis* Willd. ex Schltdl. | Child anti-oral candidiasis. Anti-asthma | Leaf | Poultice. Decoction |
| Elaeocarpaceae | *Elaeocarpus alnifolius* Baker | Anti-flu | Leaf | Decoction |
| Ericaceae | *Erica* sp. | Anti-fever | Leafy branch | Decoction |
| Euphorbiaceae | *Croton noronhae* Baill | Antidiarrheal, Anti-fatigue | Leaf | Decoction |
| Fabaceae | *Cynometra capuronii* Du Puy & R.Rabev. | Anti-yellow fever | Leaf | Decoction |
| | *Dialium unifoliolatum* Capuron | Anti-stomach ache | Leaf | Decoction |
| Gentianaceae | *Anthocleista madagascariensis* Baker | Antidiarrheal | Leaf | Decoction |
| | *Tachiadenus carinatus* (Desr.) Griseb. | Aerial part: Anti-fever Root: deworming | Leaf, stem, root | Decoction |

**Table 3.** *Cont.*

| Family | Taxon | Healing Properties | Part Used | Method of Preparation |
|---|---|---|---|---|
| Hypericaceae | *Psorospermum chionanthifolium* Spach | Antidiarrheal | Leaf | Decoction |
| Melastomataceae | *Medinilla parvifolia* Baker | Anti-prolonged cough for adults | Leaf | Decoction |
| | *Medinilla quadrangularis* Jum. & H. Perrier | Anti-prolonged cough for adults | Leaf | Decoction |
| Menispermaceae | *Burasaia madagascariensis* DC. | Anti-fatigue against hernia face mask (masonjoany). | Bark | Decoction, Poultice |
| | *Tinospora* sp. | Invigorating; anti-stomach ache; against hernia | Stem | Decoction |
| Myristicaceae | *Brochoneura acuminata* (Lam.) Warb. | Child anti-oral candidiasis; anti-stomach ache | Bark, latex | Poultice |
| Ochnaceae | *Campylospermum obtusifolium* (DC.) Tiegh. | Teeth care | Bark | Poultice |
| Olacaceae | *Olax emirnensis* Baker | Limitation of severe bleeding during delivery; anti-flu | Bark | Decoction |
| | *Noronhia boivinii* Dubard | Anti-fatigue; against swelling of the feet | Leaf | Decoction |
| Phyllanthaceae | *Bridelia tulasneana* Baill. | Anti-yellow fever, anti-oedema, Dietetic | Leaf, stem | Decoction |
| | *Uapaca thouarsii* Baill. | Aphrodisiac | prop roots | Decoction |
| Physenaceae | *Physena madagascariensis* Steud. | Antidote emetic; anti-stomach ache | Leaf | Decoction |
| Pittosporaceae | *Pittosporum ochrosiifolium* Bojer | Against eye infection | Leaf | Poultice, Infusion |
| Putranjivaceae | *Drypetes madagascariensis* (Lam.) Humbert & Leandri | Revitalizing | Leaf | Decoction |
| Rhamnaceae | *Bathiorhamnus louvelii* (H.Perrier) Capuron | Anti-fever; antidiarrheal. | Root | Decoction |
| Rhizophoraceae | *Macarisia pyramidata* Thouars | Antidiarrheal | Leaf | Decoction |
| Rubiaceae | *Breonia madagascariensis* A.Rich. ex DC. | Against toothache | Latex | Poultice |
| | *Pyrostria major* (A.Rich. ex DC.) Cavaco | Used for abortion | Bark, Leaf | Decoction, Infusion |
| | *Saldinia axillaris* (Lam. ex Poir.) Bremek. | Anti-stomach ache | Leaf | Decoction |
| Salicaceae | *Ludia madagascariensis* Clos | Anti-hemorrhagic Anti-fatigue | Leaf | Decoction |
| Sapindaceae | *Pseudopteris decipiens* Baill. | Antidiarrheal; Anti-stomach ache | Leaf | Decoction |

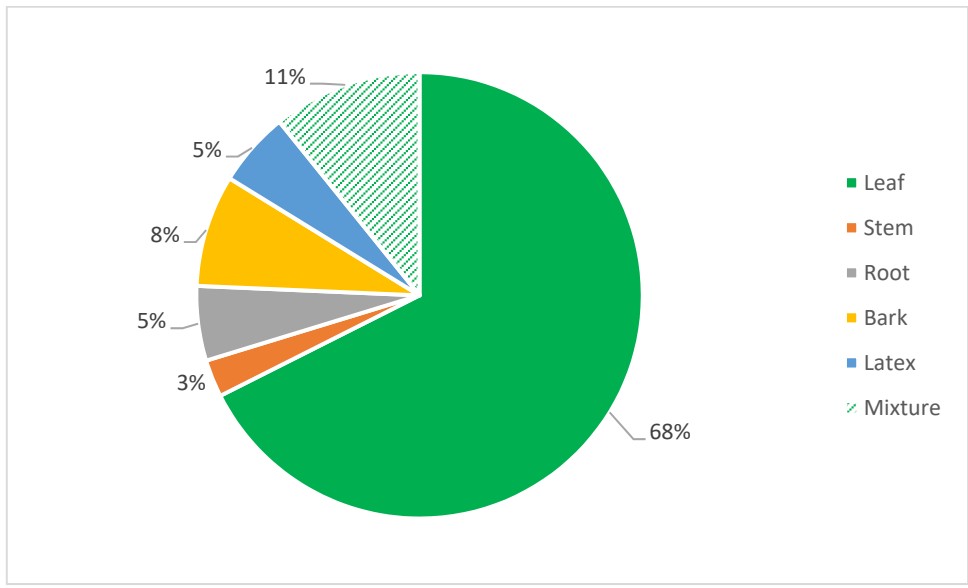

**Figure 4.** Graphic representation of plant parts frequently used for treatment.

*3.5. Phytochemical Screening for Medicinal Plants*

Twenty taxa out of the 43 medicinal plants were selected for the phytochemical analysis due to the cost and time; and also some of the plant were rained on while in the field and they got damp being destroyed by fungal attack. Leaves of 20 taxa were used for the phytochemical analysis (Tables 4 and 5). The result revealed that polyphenols, deoxy-sugar, steroids, and unsaturated sterols were the most frequently present in the analyzed medicinal plants. In contrast, alkaloids, iridoids, and flavonoids were only present in a few species.

**Table 4.** Secondary metabolites present in the leaf samples. (+: Present; −: Absent; ±: Trace).

| Species | Alkaloid | Polyphenols | Catechic Tannins | Gallotannins | Saponin | Iridoid | Deoxy-Sugar |
|---|---|---|---|---|---|---|---|
| *Brexia madagascariensis* (Lam.) Thouars ex Ker Gawl. | − | ± | + | − | + | ± | − |
| *Bridelia tulasneana* Baill. | − | + | + | + | + | − | + |
| *Brochoneura acuminata* (Lam.) Warb. | − | + | − | + | + | − | + |
| *Burasaia madagascariensis* DC. | + | + | + | − | − | − | + |
| *Cynometra capuronii* Du Puy & R.Rabev | − | + | − | + | + | − | − |
| *Dialium unifoliolatum* Capuron | + | + | − | − | − | − | − |
| *Drypetes madagascariensis* (Lam.) Humbert & Leandri | − | + | + | − | + | − | − |
| *Elaeocarpus alnifolius* Baker | − | + | − | − | − | − | − |
| *Ludia madagascariensis* Clos | − | + | + | + | − | − | + |
| *Macarisia pyramidata* Thouars | − | + | − | + | − | − | ± |
| *Noronhia boivinii* Dubard | − | + | − | − | − | − | − |
| *Olax emirnensis* Baker | − | ± | − | − | − | − | + |
| *Physena madagascariensis* Steud. | + | + | + | − | + | − | − |
| *Pittosporum ochrosiifolium* Bojer | − | + | + | − | + | − | − |
| *Pseudopteris decipiens* Baill. | − | + | − | − | − | − | ± |
| *Pyrostria major* (A.Rich. ex DC.) Cavaco | − | + | + | − | − | − | − |
| *Saldinia axillaris* (Lam. ex Poir.) Bremek. | − | + | + | + | − | + | − |
| *Secamone obovata* Decne. | − | + | + | + | − | − | − |
| *Tachiadenus carinatus* (Desr.) Griseb. | − | ± | − | − | + | − | − |
| *Tetracera madagascariensis* Willd. ex Schltdl. | − | + | + | + | − | − | ± |

**Table 5.** Secondary metabolites present in the leaf samples. (+: Present; −: Absent; ±: Trace) (*Continuation of* Table 4).

| Species | Anthraquinone | Flavonoid | Leucoanthocyanins | Steroid(s) | Triterpene | Unsaturated Sterols |
|---|---|---|---|---|---|---|
| *Brexia madagascariensis* (Lam.) Thouars ex Ker Gawl. | − | − | + | + | + | + |
| *Bridelia tulasneana* Baill. | − | − | − | + | + | + |
| *Brochoneura acuminata* (Lam.) Warb. | + | − | + | − | + | + |
| *Burasaia madagascariensis* DC. | − | − | − | + | + | + |
| *Cynometra capuronii* Du Puy & R.Rabev. | ± | − | + | + | − | − |
| *Dialium unifoliolatum* Capuron | − | + | − | + | + | + |
| *Drypetes madagascariensis* (Lam.) Humbert & Leandri | + | − | − | + | − | − |
| *Elaeocarpus alnifolius* Baker | + | − | + | + | − | + |
| *Ludia madagascariensis* Clos | + | − | − | + | − | − |
| *Macarisia pyramidata* Thouars | + | − | + | + | + | + |
| *Noronhia boivinii* Dubard | − | − | − | + | + | + |
| *Olax emirnensis* Baker | − | − | + | + | + | − |
| *Physena madagascariensis* Steud. | − | − | − | + | − | − |
| *Pittosporum ochrosiifolium* Bojer | − | − | − | + | + | + |
| *Pseudopteris decipiens* Baill. | − | − | − | + | − | + |
| *Pyrostria major* (A.Rich. ex DC.) Cavaco | − | − | + | + | − | + |
| *Saldinia axillaris* (Lam. ex Poir.) Bremek. | − | − | − | + | − | − |
| *Secamone obovata* Decne | − | − | + | + | + | + |
| *Tachiadenus carinatus* (Desr.) Griseb. | − | − | + | + | + | + |
| *Tetracera madagascariensis* Willd. ex Schltdl. | − | − | + | + | + | + |

### 3.6. Conservation (IUCN) Status

The conservation status of most of the taxa (41%) collected from Tampolo forest were of least concern (Figure 5). However, they were recorded to show a decreasing trend with few having a stable population. This was followed by vulnerable species (VU), and near threatened (NT), which recorded 11% and 8%, respectively. The endangered taxa were three, which included *Cynometra capuronii* Du Puy & R.Rabev. *Memecylon thouarsianum* Naudin, and *Cleistanthus capuronii* Leandri, whereas one taxa, *Breonia madagascariensis* A.Rich. ex DC. was recorded which is crtically endangered (CR). A total of 22% of the species were not assessed for their conservation status, and 14% of the taxa we could not identify to the species level, hence we could not assess their conservation status, and thus recorded them under unknown section.

The major threats to the taxa recorded were biological resource use, agriculture and aquaculture, and natural system modifications (Figure 6). Human intrusions and disturbances and invasive and other problematic species, genes, and diseases were recorded to be of the least threat to the taxa.

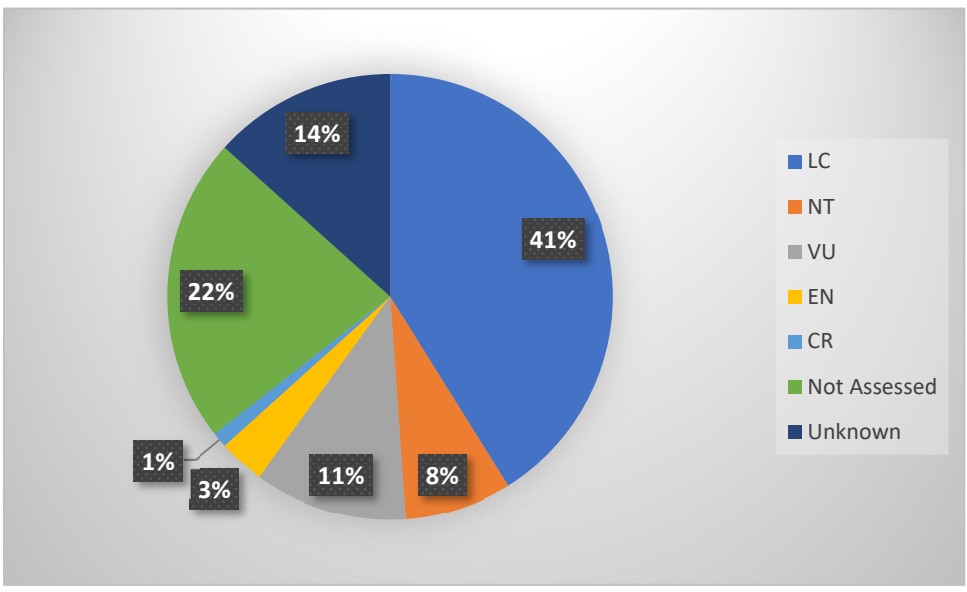

**Figure 5.** The conservation status of the taxa collected from Tampolo forest basing on the IUCN database: LC- Least Concern; VU- Vulnerable; NT- Near Threatened; CR- Critically Endangered; EN- Endangered.

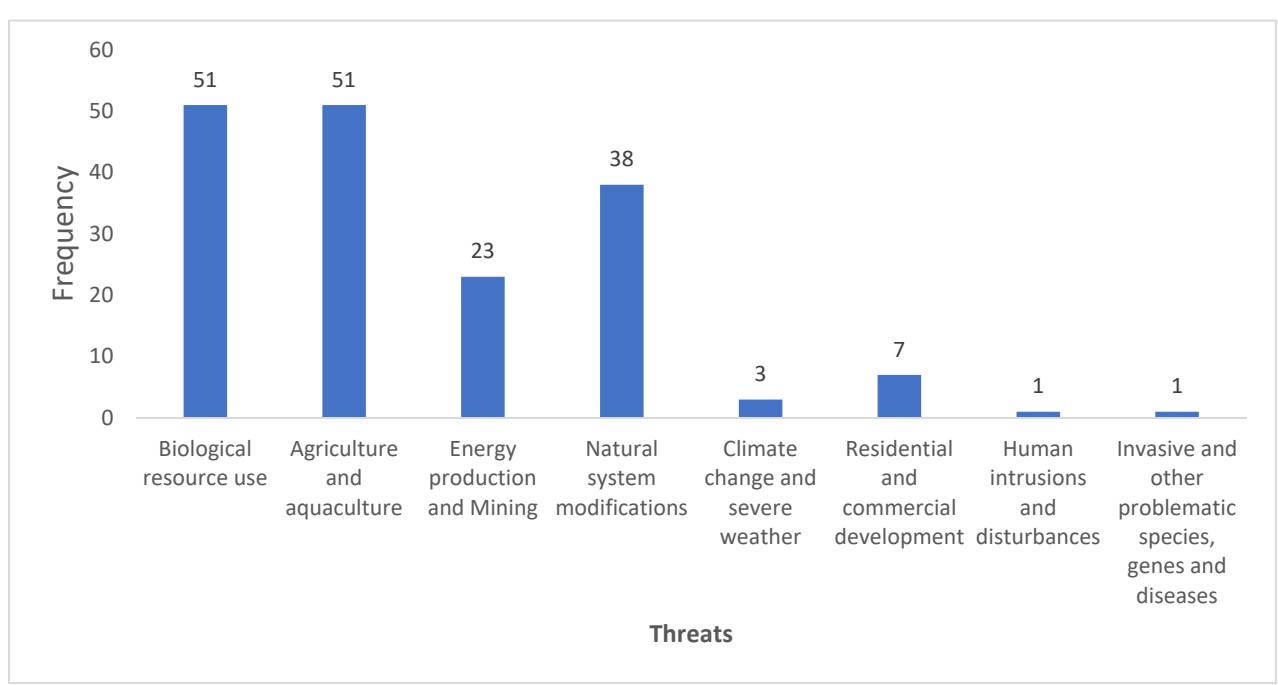

**Figure 6.** The most recorded threats from the IUCN database of the plant taxa.

### 4. Discussion

All the informants who participated in this study were over 25 years old, mostly dwelling in Tampolo which is a rural area. Rural communities have been known to utilize the natural resources to satisfy their daily needs [56]. This is due to the low income, lack of alternative sources of income, and lack of modern healthcare facilities within the regions [57,58]. It was observed in the survey that their knowledge of useful plants, mostly medicinal plants, was passed down from their ancestors through oral traditions. Preliminary engagements have shown that younger people are ignorant of traditional knowledge; in contrast, the elderly interviewees depend mostly or entirely on natural forest resources.

Despite its small size of about 1/6500 (0.015%) of the total cover the Malagasy Rainforests, the flora of Tampolo forest is highly diverse having 360 plant taxa [30], which represents 2.6% of the flora of Madagascar [3]. The assumption that local communities use forest plant species as timber, firewood and especially for medicinal purpose was demonstrated in this work. This study demonstrated most of the plant taxa found in this region were useful in the treatment of various illnesses. Hence, the use of traditional medicine is an important part of the healthcare of the Tampolo Community. Their dependence on natural resources for their livelihood and their basic healthcare were due to their economic state, lack of health facilities in the remote regions of the country, and their socio-cultural situation [59–61]. These species are utilized by the local adjacent communities to fulfill their daily livelihood needs [9,12,15,62,63].

Our findings were similar with other botanical surveys that showed the importance of forest species to the local communities [13]. The conservation status of most of the plant taxa were of least concern, however, from the IUCN database, they were shown to have a decreasing trend. Hence, protection of these forests where they occur is critical, in addition to awareness to the researchers and policy-makers and also the local communities. Non-severe health problems, such as fever and digestive disorders, were the most commonly treated with medicinal plants. Similarity in the mode of use and the recorded healing properties of several species were observed in different areas across Madagascar [64]. For example, leaves of *Phyllarthron bojeranum* DC (Bignoniaceae) were also used as treatment of fatigue in Analangazaha Farafangana [26,65], in Ambalabe Vatomandry [21], in Antananarivo [25,66] and in Tampolo. This is also the case of *Anthocleista madagascariensis* Baker which were an antidiarrheal [25]. Phytochemical analysis revealed the presence of active secondary metabolites which have been linked to the treatment of various diseases [67] in the 20 selected forest taxa. However, more studies need to be undertaken to test their efficacy. The healing properties of the medicinal plants are, in part, due to the presence of the secondary metabolites such as alkaloids, saponins, flavonoids, tannins, glycosides, anthraquinones, steroids, and terpenoids [68]. Phytochemical screening was used to detect the presence of them following the procedure of Cordell [41], Hemingway and Karchesky [42], and Bruneton [40]. Some secondary metabolites, such as phenols, tannins, flavonoids, and quinones, show antidiarrheal effects [69–72]. Compounds such as alkaloids, phenols, tannins, iridoids, flavonoids, steroids, and terpenes have been shown to have anti-inflammatory, antioxidant, antiseptic properties [69–74]. Based on these previous literatures, their presence can justify the specified therapeutic properties of the plant. In this study, *Saldinia axillaris* (Lam. ex Poir.) Bremek., indicated as an antidiarrheal, contains polyphenols, and *Tetracera madagascariensis* Willd. ex Schltdl., used to treat oral candidiasis, contains polyphenols and tannins which are antiseptics. However, further analysis should be done to prove an indepth understanding into their efficacy.

Medicinal plant gathering by the local people is generally non-destructive because the quantity of the collected leaves is relatively small and only used for daily dose and family use. The same holds true for the need of firewood because only dead wood can be collected and that is under the control of the protected area managers. Moreover, although most of the population did not go past elementary school in terms of their education level, awareness campaigns have been implemented by the ESSA-Forêt (Ecole Supérieure des Sciences Agronomiques-Forêt) and their partners, allowing the raising of awareness of these people of the ecological, environmental, and socio-economic importance of the biodiversity that Tampolo forest shelters and that their participation in conservation acts have been noticed. However, due to the increasing demand for wood products, exploitation of the forest species focused more on the timber harvesting rather than the medicinal uses or the edibles, essentially to feed the markets of certain cities, such as Fenoarivo, Atsinanana, and Ampasina Maningory, promoting the non-selective and illegal logging which worsen the pressures weighing heavily on the protected area. In addition, the lack of written documents from the herbalists and the traditional healers [20], and the lack of interest from young generation to the tradition, has led to the decrease of traditional medicine and

medicinal plant knowledge. This loss of knowledge was reported by Ravelonanosy in 2018 [63], while only 53 medicinal species were documented instead of 59 in 2012.

## 5. Conclusions

This survey generally revealed that most of the plant taxa from the Tampolo forest were mostly utilized for medicinal purposes and for timber by the adjacent local community. A total of 84.8% of these useful plant taxa are endemic to the Madagascar region. This documentation of ethnobotanical knowledge provides a catalog of useful plants of the Tampolo, and will serve as a physical record of their culture for the education of future generations. It will also strengthen their culture by recognizing their traditional knowledge on medicinal plants and providing scientific basis for it. However, the overexploitation may disturb the ecological balance of the area which subsequently can lead to the disappearance of these species. Hence, further efforts on environmental education still should be provided because Tampolo is one of the last remnant littoral forests of East Madagascar, thus, this could help conserve this area/forest. Necessary measures should also be taken to protect these most exploited species to avoid their future extinction. The current finding can be used as a reference point for various studies within the forest to help reconcile the local livelihood needs with forest conservation. Based on the findings, we recommend further studies regarding ecology, conservation, and chemistry of the remaining species which constitute the flora of the littoral forest of Tampolo.

**Author Contributions:** G.E.O. conducted the interviews and completed the data analysis. C.S. and M.B.R. offered technical support in the field, C.S., V.H.J. and G.H. supervised the work, and gave constructive comments. V.O.W., E.M.M., J.K.M. performed and reviewed the analyzed data. B.M.R.R. drew the map. All authors have read and agreed to the published version of the manuscript.

**Funding:** This work was financially supported by the Association de Valorisation de l'Ethnopharmacologie en Régions Tropicales et Méditerranéennes (AVERTEM), University of Lilles, France, and grants from National Natural Science Foundation of China (31970211) and Sino-Africa Joint Research Center, CAS, China (SAJC202101).

**Institutional Review Board Statement:** Not applicable.

**Informed Consent Statement:** Informed consent was obtained from all subjects involved in the study.

**Acknowledgments:** The authors would like to thank the Association de Valorisation de l'Ethnopharmacologie en Régions Tropicales et Méditerranéennes (AVERTEM) for the financial and technical support of this study and the Department of Plant Biology and Ecology Ankatso Antananarivo to allow them to carry out this research. We also sincerely thank Ministère de l'Environnement de du Développement Durable for issuance of the permits for the research. Finally, they would like to thank all of the people who contributed to the elaboration of this study.

**Conflicts of Interest:** The authors declare no conflict of interest. The funders had no role in the design of the study; in the collection, analyses, or interpretation of data; in the writing of the manuscript, or in the decision to publish the results.

## Appendix A

Questionnaire for conducting the ethnobotanical survey on useful plants of the tampolo forest.

(A) Information on the person to be investigated

- Name:
- Age:
- Gender:
- Profession:

(B) Questionnaire

- Do you use plants for healing?
- If yes, do you know of plants to treat such a disease? (For example, diarrhea).

- How do you use this plant? and how much and what part?
- And for other diseases, which plants do you use? and how?
- How much are you taking and for how long?
- How do you recognize that such a plant has medicinal properties?
- Where do you find these plants? Are you planting them or picking them somewhere? If this is the second case, where?
- What are some the medicinal forest species?
- What other species do you collect from the forest? What parts and
- What quantities?
- Is it for your own use that you collect these plants or for business purposes?
- What are the criteria for recognizing each plant? Does his name have anything to do with this? criterion?
- Have you thought of ways to prevent these plants from going extinct?

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
