# Peer review of "Ethnobotanical Survey in Tampolo Forest (Fenoarivo Atsinanana, Northeastern Madagascar)"

_forests, doi:10.3390/f12050566_

Round 1

Reviewer 1 Report

The authors conducted a very interesting study. In general, I like the manuscript and would make comments that would improve the general quality of the manuscript.

The main comments stem from the methodology. I feel that authors should clearly state how the respondents were selected, and the reason for selecting only 20 plant species for the phytochemical screening, out of the numerous plants they documented. Also, in Table 2, I suggest authors search through the IUCN and include the conservation status for each of the species. Then, from the IUCN red list, they can have an analysis showing how many species are the least concern and threatened. They could also check the other threats affecting each of the species, and then do a detailed threat evaluation. Including these suggestions would boast the general quality of the paper. 

Again, the authors provide the questionnaire used for gathering the information. This should be included as a supplementary file in this manuscript.

For other corrections, please see the attached file.

Author Response

Thank you so much for the comments and suggestions. We have tried very much to include them in the main manuscript. please find the attached document for the responses.

Reviewer 2 Report

Onjalalaina and colleagues present results of an ethnobotanical survey of a forest in northeastern Madagascar. Since deforestation of Madagascar continues at an enormos speed and many species in this biodiversity hotspot are highly threatened, this is an important task.

In general, the study design and method is straight forward and the results are clear. My main concern is that contrary to the statement in l. 126, the Nagoya protocol seems to have been ignored. Obviously, the study was led by a Chinese institute and partly funded by a French association but no Material transfer agreements, no Research or collecting permit numbers are given. If these don't exist, the study cannot be published since this would be a violation of Malagasy law. The statement of l. 121/122 is strange: universities don't give research permits, they have to obtain them from governmental organisations.

Besides this, I have only minor comments:

  • title: maybe better to write "northeastern Madagascar" since most readers will not be able to place Fenoarivo on a map (?).
  • l. 55 - "a further loss of 4,345,000 hectares"? This would mean that no forest is left?
  • l. 59 - delete "mostly"
  • l. 63 - since the paper is about plants, mentioning the Aye-Aye here is a bit confusing
  • l. 121 - and the government? permit numbers?
  • l. 126 - in what respect has the Nagoya protocol been followed? To me, this looks like ethnobotanical prospection by a Chinese lab without any benefits for Madagascar. I might be wrong - but the information to show that I am wrong is missing (material transfer agreements or any other benefit sharing agreements??)
  • l. 139 - Please add table of voucher specimens with voucher numbers and herbarium acronyms.
  • l. 196ff - Most of this paragraph is copied from the intro (l. 112 ff.). The authors should decide if this information is better in the intro or in the results but no need to duplicate it.
  • Table 1 - title: for "repartition" read "distribution"
  • P. 7, l. 12 - for "Decne" read "Decne."
  • l. 296 - this is somewhat a contradiction to the statement in l. 288/289 that only dead wood is collected and this activity is controlled by the rangers

Author Response

Thank you very much for the comments and suggestions. Please see the attachment

Reviewer 3 Report

Dear Authors,

I have carefully read your manuscript titled "Ethnobotanical survey in Tampolo forest (Fenoarivo Atsinanana, Madagascar)" and I found it very interesting for the novelties reported on useful plants. However, several aspects must be checked by you.

Methodology for the phytochemical screening for medicinal plants it seems not so updated. Please, check and revise some more recent phytochemestry technics.

In general, in the text and in tables and figures, it is better to indicate the botanical entities as "taxon" or "taxa".

In scientific nomenclature, the dots at the end of some author name is not an optional. Please, check and revise all the scientific names in your text and tables using a matching tool on GBIF.org or other similar website consulted.

Finally, check and correct the italics form of the scientific nomenclature in the References section.

Other notes, suggestions and corrections are reported in the attached PDF.

Best wishes.

Author Response

Thank you very much for the comments and responses. Please see the attachment

Round 2

Reviewer 3 Report

Dear Authors,

thanks to the suggestions of the three reviewers you improved very well your manuscript. I pointed out just two little highlighted parts in the attached PDF for a proper correction.

Best regards.
